# Effects of DA-9701 (motilitone®) on gastric emptying, glycemic control, and oxidative stress in diabetic rats

**Minjun Kim** [1], **Young Mi Song** [2], **Kyung Jae Yoon** [3], **Yurim Hong** [2], **Jung Ho Park** [4], **Yoo Kyung Kim** [4], **Juhee Kim** [2], **Chong Il Sohn** [4]*

1 Department of Gastroenterology, Chung-Ang University Gwangmyeong Hospital, Chung-Ang University School of Medicine, Gyeonggi-do, Republic of Korea, 2 Institute of Medical Research, Samsung Kangbuk Hospital, Sungkyunkwan University School of Medicine, Seoul, Republic of Korea, 3 Department of Rehabilitation Medicine, Samsung Kangbuk Hospital, Sungkyunkwan University School of Medicine, Seoul, Republic of Korea, 4 Department of Gastroenterology, Samsung Kangbuk Hospital, Sungkyunkwan University School of Medicine, Seoul, Republic of Korea

☯ These authors contributed equally to this work.
* Chongil.sohn@samsung.com

## Abstract

### Background/aims

Diabetes is associated with various gastrointestinal disorders, including altered gastric emptying, which may be rapid, slow, or transient. These alterations can significantly influence gastrointestinal symptoms and directly influence blood glucose levels. DA-9701 (motilitone), a prokinetic agent derived from *Corydalis* tuber and Pharbitis seed, is employed in Korea to manage functional dyspepsia due to its anti-inflammatory and gastrointestinal motility-enhancing properties. This study aims to investigate the potential of DA-9701 in addressing altered gastric emptying and glycemic control in diabetic rats, thereby validating its broader clinical utility.

### Materials/methods

Diabetes mellitus was induced in rats by streptozocin injection (65 mg/kg, i.p.). Following the onset of diabetes, rats received daily oral administration of DA-9701 for 2 weeks. Gastric emptying rates for liquid and solid meals were measured using plasma acetaminophen levels and residual food mass, respectively. Oral glucose tolerance tests (OGTT), insulin levels, and oxidative stress markers (malondialdehyde [MDA], Ogg1, Gpx, Cat) were assessed. Western blotting and qPCR were used to evaluate the expression of ERK1/2, c-Kit, and proliferating cell nuclear antigen (PCNA) in gastric tissue.

### Results

Diabetic rats exhibited significantly accelerated gastric emptying (liquid GE AUC: +45.2%; solid GE: +23.1%, $p < 0.01$) and elevated blood glucose

**Data availability statement:** All relevant data are within the paper and its Supporting Information files.

**Funding:** This work was supported by the National Research Foundation of Korea (NRF-2020R1F1A106597511 and NRF-2021R1A2C400245413) to Dr. Chong Il Sohn.

**Competing interests:** The authors have declared that no competing interests exist.

$(327.4 \pm 22.8\,\text{mg/dL}$ vs. $96.2 \pm 10.1\,\text{mg/dL}$ in controls, $p < 0.001$), accompanied by increased oxidative stress markers and expression of c-Kit, ERK1/2, and PCNA. DA-9701 treatment normalized gastric emptying rates (solid GE restored to 55.8%, $p < 0.05$), reduced MDA, Ogg1, Gpx, and Cat expression, and significantly down-regulated ERK1/2, c-Kit, and PCNA. Moreover, insulin secretion increased 2.1-fold in DA-9701-treated diabetic rats ($p < 0.05$), resulting in improved glucose tolerance (OGTT AUC reduction: $-24.6\%$, $p < 0.01$).

## Conclusion

DA-9701 normalized gastric emptying and glycemic control while reducing the expression of ERK1/2, c-Kit, and PCNA, which are elevated in diabetic gastric tissues. These findings highlight the dual therapeutic potential of DA-9701 in regulating both gastrointestinal motility and glycemic variability in diabetes, warranting further investigation in clinical settings.

## Introduction

Diabetes mellitus (DM) is a prevalent chronic metabolic disorder that disrupts metabolic equilibrium and often leads to overlooked complications, such as gastrointestinal dysfunction. A significant issue in diabetes management is abnormal gastric emptying, which manifests as dyspepsia, early satiety, nausea, and vomiting [1], complicating glycemic control and significantly impacting quality of life. Gastric emptying, influenced by neural and hormonal mechanisms, can be disrupted in diabetes, particularly due to autonomic neuropathy and neurological abnormalities [2,3]. These disruptions exacerbate blood glucose irregularities, further complicating diabetes management [4].

At the cellular level, diabetes affects the expression of neuronal nitric oxide synthase and the functionality of interstitial cells of Cajal (ICCs), which are crucial for gastrointestinal motility [5]. Changes in ICCs highlight the importance of the enteric nervous system and intercellular communication within the gastrointestinal tract [6].

While most studies have focused on delayed gastric emptying [6–8], some have reported accelerated gastric emptying in a subset of patients with type 1 or type 2 DM, particularly in those with poorly controlled diabetes [4,9,10].

This study used a diabetic rat model to investigate early-stage alterations in gastric emptying and the dynamics within ICC networks. ICCs, which are essential for generating electrical slow waves and facilitating nerve–muscle interactions, are crucial for gastrointestinal motility. The pathophysiology of diabetes-related damage to ICCs is complex and may involve oxidative stress mechanisms [9].

DA-9701, known as motilitone®, is a prokinetic agent formulated from *Corydalis* tuber and *Pharbitis* semen, traditionally used in Oriental medicine for gastrointestinal issues [11]. It enhances gastric emptying and accommodation by acting on various receptors, including D2 and 5-HT4. The components of DA-9701, such as corydaline and chlorogenic acid, contribute to its analgesic, antiulcer, and prokinetic effects. Due

to its natural origins, this botanical drug requires complex quality control to ensure consistency and efficacy in treating functional dyspepsia and other gastrointestinal disorders [11,12].

In this study, we investigated the early alterations in gastric emptying in streptozocin (STZ)-induced diabetic rats and assessed the potential mitigating effects of DA-9701 treatment. We also evaluated the impact of DA-9701 on blood glucose levels in relation to diabetes-induced changes in gastric emptying. This research aims to provide insights into the therapeutic potential of DA-9701 in addressing diabetes-induced gastrointestinal complications, highlighting the need for holistic treatment approaches in diabetic patients.

## Materials and methods

### Animals and induction of diabetes

Six-week-old male Sprague Dawley rats were purchased from DLB in South Korea. The animals were maintained under controlled conditions (21°C±2°C; 60%±10% humidity; 12-hour light/dark cycle) with free access to food and water. The rats were acclimatized to these conditions for 7 days and then received a single intraperitoneal injection of freshly prepared STZ (65 mg/kg body weight) dissolved in 10 mM citrate buffer (pH 4.5). Once hyperglycemia was confirmed, rats with a non-fasting blood glucose level >250 mg/dL were kept on standby for at least 2 weeks before being used in the experiment. Body weight and blood glucose levels were measured weekly (Fig 1). As shown in Fig 3, to assess changes in liquid and solid gastric emptying over the duration of diabetes induction, rats were divided into four groups based on age, using STZ-induced diabetic and age-matched control rats: (1) normal vs. diabetic group (n = 5 per group) at 2 weeks after injection with vehicle or STZ; (2) normal vs diabetic group (n = 4–6 per group) at 4 weeks after injection with vehicle or STZ. All the rats were euthanized using 100% carbon dioxide, and pancreatic weight was measured. All animal experiments complied with the Institutional Animal Care and Use Committee guidelines (approval number 202007130) and adhered to ARRIVE guidelines.

### Oral glucose tolerance

Rats were fasted overnight (approximately 16 hours), and oral glucose tolerance tests (OGTTs) were conducted using 1 g/kg body weight of glucose (10% solution). For blood glucose level assessment, a single drop of blood obtained via tail vein puncture was placed on a glucometer strip (Roche Diagnostics, Basel, Switzerland). Following animal protocol guidelines for blood volume collection per rat, blood samples were collected at 0, 15, 30, and 60 minutes after glucose administration using heparin-coated capillary microcuvette tubes (SARSTEDT AG & Co. KG, Nümbrecht, Germany) to measure insulin levels. The blood collection tubes were centrifuged at 13,000 rpm for 5 minutes at 4°C to isolate plasma, which was then stored at −80°C until further analysis.

### Liquid gastric emptying tests

To assess the gastric emptying rate for liquids, all rats were fasted overnight (approximately 16 hours) and subsequently administered a 10% (w/v) glucose solution (1 g/kg body weight) containing 1% (w/v) acetaminophen (100 mg/kg body weight) orally. Blood samples were taken from the tail vein at 0, 15, 30, and 60 minutes [13,14]. Acetaminophen levels in the plasma were measured using an enzymatic assay kit (Cambridge Life Sciences Ltd, Ely, Cambridgeshire, UK).

### Solid gastric emptying tests

Fasted rats were placed in individual cages with *ad libitum* access to water and a preweighed amount of standard rat chow (5 g) for 30 minutes. After feeding, the collected spillage was weighed to accurately measure food consumption. Rats were euthanized four hours after eating. The stomach was exposed via laparotomy. The pyloric junction was ligated, and the stomach was removed. The stomach contents were extracted and dried according to a previously established

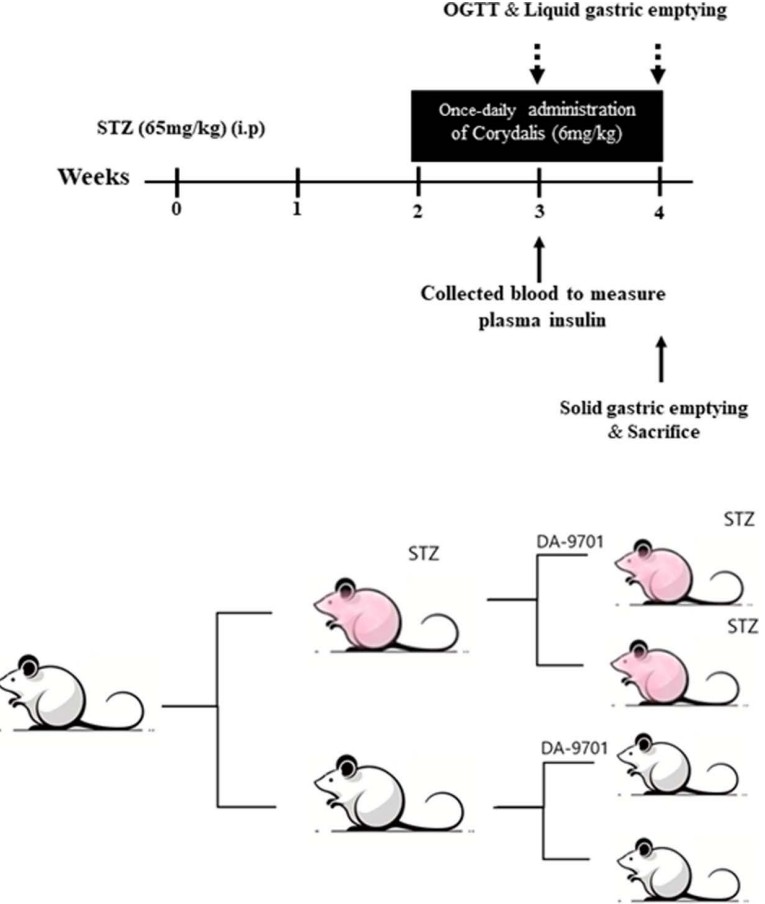

**Fig 1. Schema of experimental schedule.** Experimental protocol used to induce diabetes with streptozocin (STZ). All rats were administered daily with DA-9701 or vehicle by oral gavage for 2 weeks. Oral glucose tolerance tests (OGTT) and liquid gastric emptying assessments were performed at weeks 3 and 4. The study was completed with solid gastric emptying measurements after sacrifice of the rats.

protocol to determine dry matter content. Solid gastric emptying was calculated using the following formula: gastric empty-ing (%) = {1 − [dry matter of stomach contents (g)]/[dry matter of food intake (g)]} × 100 [15].

## Effects of DA-9701

The active pharmaceutical ingredient (API) used in this study was DA-9701 (Motilitone), a proprietary pharmaceutical product provided by Dong-A ST (Seoul, South Korea). DA-9701 is a standardized combination of *Corydalis* tuber extract and *Pharbitis* semen extract, formulated at a 5:1 ratio. The extracts were prepared and standardized under strict quality control guidelines in compliance with Good Manufacturing Practice (GMP) standards. The *Corydalis* tuber and *Pharbitis* semen extracts used in DA-9701 were supplied by Daepyung Co., Ltd. (Gyeongsangbuk-do, South Korea). The man-ufacturing process included freeze-drying of the extracts, achieving a final yield of 93.5%, as verified by a Certificate of Analysis (provided as supplementary material). The chemical characterization of the extracts included the quantification of active compounds such as Corydaline and Imperatorin, as well as evaluations for impurities, heavy metals, and microbial contamination, ensuring the extracts meet pharmaceutical-grade quality requirements.

Diabetic rats and their age-matched controls were randomly divided into four groups according to treatment: group 1–normal (rats with normal glucose levels and received sodium citrate buffer vehicle) + 5% cellulose; group 2–normal + DA-9701; group 3–DM (diabetic rats with high blood glucose levels) + 5% cellulose; and group 4–DM + DA-9701 (6 mg/kg for 2 weeks). The rats were orally administered daily with DA-9701 (6 mg/kg) or 5% cellulose for 2 weeks.

### RNA extraction and qRT-PCR

Total RNA was extracted from the antrum using TRIzol reagent (Thermo Fisher Scientific, Waltham, Massachusetts, USA). Complementary DNA (cDNA) was synthesized from 1 µg of total RNA using the Applied Biosystems™ High-Capacity RNA-to-cDNA™ Kit (Thermo Fisher Scientific) according to the manufacturer's instructions. The cDNA from the antrum was utilized for RT-PCR with the SensiFAST™ SYBR® Lo-ROX Kit (Meridian Life Science, Memphis, Tennessee, USA) and performed on a LightCycler® 480 system (Roche Diagnostics) to determine relative mRNA levels. The primer sequences used for rat samples were as follows: *Ogg1* (8-oxoguanine DNA glycosylase-1), forward primer 5′-CTA AGA AGA CAG AAG GCT AGG TAG-3′ and reverse primer 5′-TGA CTT TGA TTT GGG ATG TTT GC-3′; *Gpx* (glutathione peroxidase), forward prim 5′- TAG GTC CAG ACG GTG TTC CA-3′ and reverse primer 5- CCT TAG GGG TTG CTA GGC TG-3′; Cat (catalase), forward primer 5′- GCT CCG CAA TCC TAC ACC AT-3′ and reverse primer 5′- GTG GTC AGG ACA TCG GGT TT-3′; *Actin*, forward primer 5′-CGT GCG TGA CAT TAA AGA G-3′ and reverse primer 5′-TTG CCG ATA GTG ATG ACC T-3′.

### ELISA for insulin and biochemical analyses

Plasma insulin and serum malondialdehyde (MDA) levels were quantified using an ELISA kit (Alpoco Diagnostics, Salem, NH, USA) and a lipid peroxidation assay kit (Abcam, Cambridge, United Kingdom), respectively, according to the manufacturers' instructions.

### Insulin measurement

we collected blood samples from mice at 0, 15, 30, and 60 minutes following glucose administration using heparin-coated capillary microvette® tubes (SARSTEDT, 20.1282.100) to measure plasma insulin levels. After centrifugation at 13,000 rpm for 5 minutes at 4°C, plasma was isolated and stored at −80°C until further analysis. Plasma insulin levels were quantified using an ELISA kit (Alpoco Diagnostics, Salem, NH, USA), according to the manufacturers' instructions.

### Immunofluorescence staining

Formalin-fixed rat stomach tissues were embedded in paraffin using established methods. The paraffin sections were prepared, blocked with 2% horse serum, and subsequently incubated overnight at 4°C with primary antibodies against c-KIT. After priamry incubation, the sections were incubated with Alexa Fluor 488-conjugated secondary antiboies (Invitrogen, A21206) for 60 min at room temperature and mounted with mounting medium (Abcam, ab104139). Images were observed using a confocal microscope (Leica, Stellaris 5).

### Immunoblot test

Rat stomachs were lysed in radioimmunoprecipitation assay buffer (Thermo Fisher Scientific) using a mini protease inhibitor cocktail (Roche Diagnostics). After centrifugation at 120,000 rpm for 20 minutes at 4°C, equal amounts of protein were separated by sodium dodecyl sulfate–polyacrylamide gel electrophoresis and electrophoretically transferred onto a polyvinylidene fluoride membrane (Bio-Rad Laboratories, Inc., Hercules, CA, USA). The membranes were blocked with 5% bovine serum albumin dissolved in Tris-buffered saline with Tween 20 (TBST) and incubated for 1 hour at room temperature. They were then incubated overnight at 4°C with primary antibodies against c-Kit (1:1000; LifeSpan BioSciences,

Inc., Seattle, WA, USA), protein kinase RNA-like endoplasmic reticulum kinase (pERK, 1:1000; Cell Signaling Technology, Inc., Danvers, MA, USA), extracellular signal-regulated kinase (ERK, 1:1000; Santa Cruz Biotechnology, Inc., Dallas, TX, USA), proliferating cell nuclear antigen (1:1000; PCNA; Sigma-Aldrich) heat shock protein 90 (HSP90; 1:1000; Santa Cruz Biotechnology, Inc., Dallas, TX, USA) and β-actin (1:1000; Santa Cruz Biotechnology). The membranes were subsequently washed three times in TBST for 5 minutes each and incubated with horseradish peroxidase-conjugated anti-rabbit or anti-mouse immunoglobulin G antibody (1:10,000; Santa Cruz Biotechnology) at room temperature for 1 hour. The blots were developed using an enhanced chemiluminescent Western blotting detection kit (Amersham, Buckinghamshire, UK).

### Statistical analysis

Data are presented as mean±standard error of the mean. Statistical significance was determined using an unpaired two-tailed t-test or one-way or two-way ANOVA with GraphPad Prism 9 (GraphPad Software, Boston, MA, USA). A P-value of <0.05 was considered statistically significant.

The complete set of raw numerical data used for statistical analyses is provided in S1 Raw data.

## Results

### Elevated OGTT in STZ-induced diabetic rats

Rats with STZ-induced diabetes exhibited significantly higher blood glucose levels compared with nondiabetic controls following an OGTT. At three and four weeks post-STZ administration, diabetic rats reached peak blood glucose levels at 30 minutes, maintained elevated levels at 60 minutes, and demonstrated a three- to four-fold increase compared with the control group (Fig 2A). Throughout the four-week experimental period, the body weight of both control and STZ-treated groups was monitored weekly. Initial measurements indicated no significant differences in body weight between the two groups. However, by the second week, a marked divergence in weight trajectories became evident. The STZ-treated group exhibited a significant reduction in body weight compared to the control group (p<0.01). This trend persisted, with the STZ-treated group maintaining significantly lower body weights through the third and fourth weeks of the study (p<0.0001 for both time points). The differences in blood glucose levels were statistically significant (p<0.001).

### Accelerated gastric emptying in STZ-induced early diabetic rats

Liquid gastric emptying rates were assessed using serum acetaminophen as a marker. Serum acetaminophen concentrations were measured at 2 and 4 weeks post-STZ administration. At week 4, control rats showed a gradual increase in serum acetaminophen levels. In contrast, this marker increased rapidly in diabetic rats, peaking at 15–30 minutes, followed by a gradual decline (Figs 3A and 3B).

The area under the curve bar graph indicated accelerated gastric emptying at 2 weeks, which became more pronounced at 4 weeks (Figs 3A and 3B). Solid gastric emptying, quantified by the remaining food in the stomach 4 hours postprandially, was also accelerated in diabetic rats at both 2 and 4 weeks (Figs 3C and 3D).

### Impact of DA-9701 on gastric emptying and glycemic control in early diabetic rats

Administration of DA-9701 to diabetic rats for 2 weeks resulted in a transition from rapid to normal patterns of liquid gastric emptying at 3 and 4 weeks (Fig 4A). Additionally, the initially rapid solid gastric emptying in diabetic rats normalized by week 4 (Fig 4B), whereas no significant changes were observed in the control group. Blood glucose levels showed more substantial improvement in the DA-9701-treated group compared with controls at 3 and 4 weeks (Figs 4C and 4D).

### Effects of DA-9701 on oxidative stress, gastric inflammation markers

Treatment with DA-9701 in STZ-induced diabetic rats resulted in significant modulation of cellular and molecular markers associated with gastric dysfunction. Immunofluorescence staining and immunoblot analysis revealed that c-Kit expression

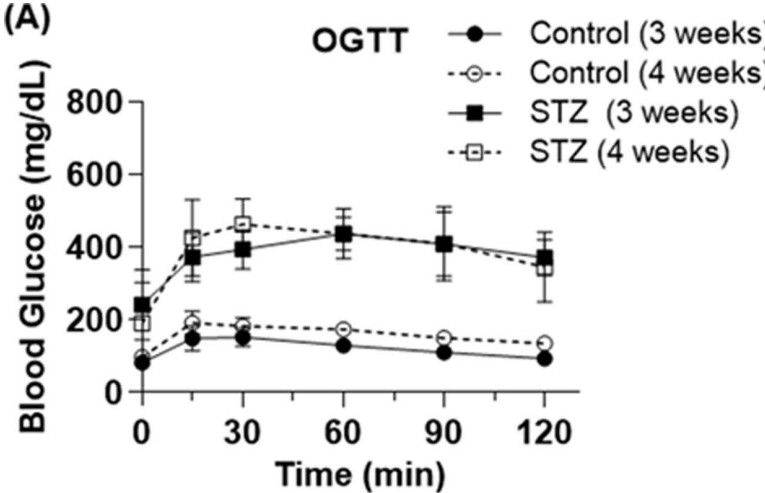

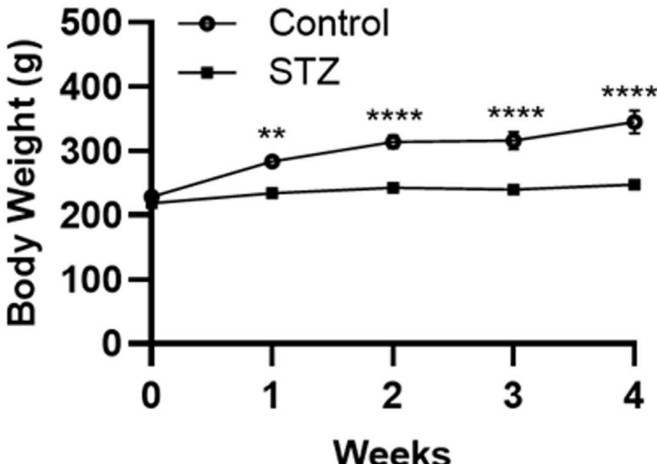

**Fig 2. Effects of STZ on OGTT and body weight in rats.** (A) OGTT at 3 and 4 weeks in control and STZ-treated rats. Control groups at 3 weeks (●) and 4 weeks (○) showed stable glucose levels post-OGTT, while STZ-treated rats at 3 weeks (■) and 4 weeks (□) exhibited elevated glucose levels. Data are shown as mean ± standard error of the mean (5–8 mice per group). *Control (3 weeks) vs. STZ (3 weeks)* – $p < 0.05$, **** - $p < 0.0001$, #-Control (4 weeks) vs. STZ (4 weeks). (B) Body weight of control and STZ-treated rats was measured weekly over a 4-week period, with the STZ-treated group showing lower body weight compared to the control group (6–7 mice per group).

was markedly increased in the gastric antrum of diabetic rats compared with controls. Administration of DA-9701 significantly downregulated c-Kit expression (Fig 5A), suggesting potential normalization of gastric motility.

Plasma malondialdehyde (MDA) levels, a marker of oxidative stress, were significantly elevated in diabetic rats and were reduced following DA-9701 treatment (Fig 5B). In addition, mRNA levels of oxidative stress-related genes, including 8-oxoguanine DNA glycosylase 1 (Ogg1), glutathione peroxidase (Gpx), and catalase (Cat), were upregulated in diabetic rats. DA-9701 administration significantly decreased the expression of these genes (*$p < 0.05$, **$p < 0.01$, ***$p < 0.001$) (Fig 5B).

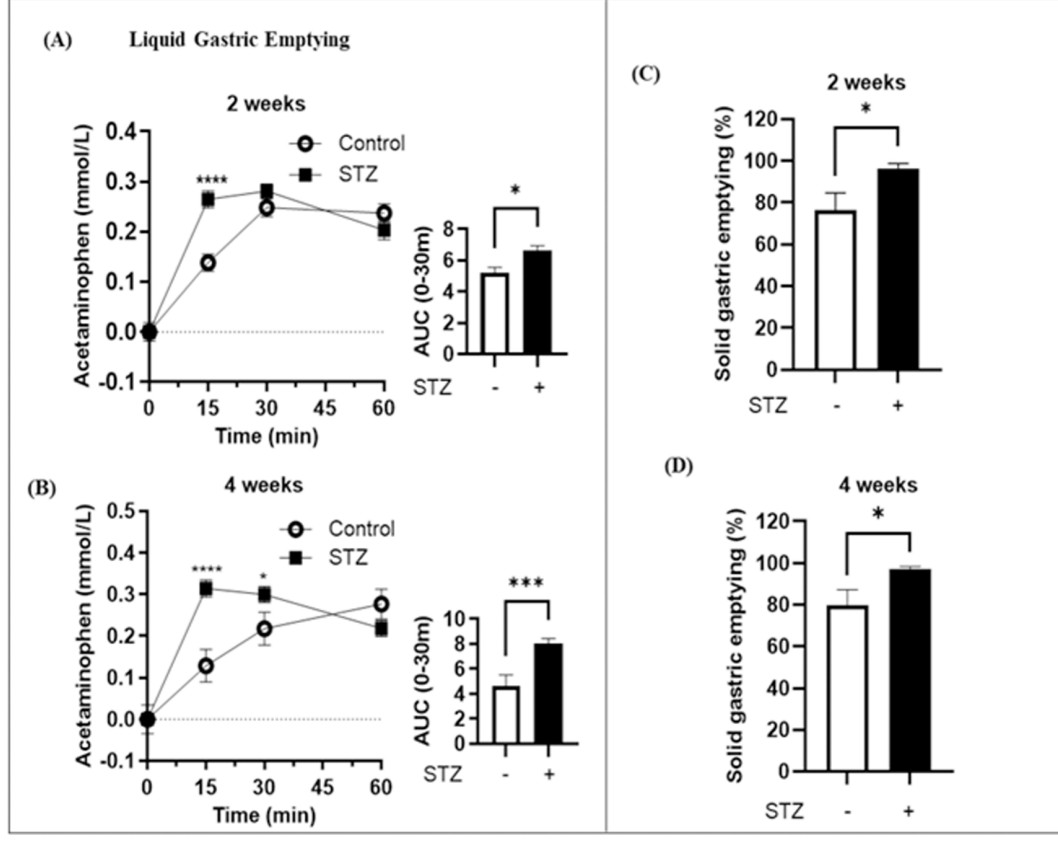

**Fig 3. Liquid and solid gastric emptying in the control and STZ-treated rats at 2 and 4 weeks.** (A) Plasma acetaminophen levels as a measure of liquid gastric emptying show increased emptying in STZ-treated rats at 2 weeks (n = 5/group; *p < 0.05). (B) Plasma acetaminophen measurement after 4 weeks indicates continued enhanced liquid gastric emptying in STZ-treated rats (n = 6–11/group; **p < 0.01, ****p < 0.0001). (C) Solid gastric emptying is significantly accelerated in STZ-treated rats at 2 weeks (*p < 0.05). (D) At 4 weeks, STZ treatment maintains faster solid gastric emptying compared to control (*p < 0.05) (n = 4–6/group).

Immunoblot analysis demonstrated that phosphorylation of ERK1/2 (p-ERK1/2) and expression of proliferating cell nuclear antigen (PCNA) were increased in the gastric tissues of diabetic rats. These increases were significantly attenuated by DA-9701 treatment (Fig 5C). All analyses were performed using gastric tissues obtained from control and diabetic rats treated with vehicle or DA-9701 (6 mg/kg) daily for two weeks.

In support of these findings, Western blot analyses confirmed the upregulated expression of c-Kit (S2 Fig), p-ERK (S3 Fig), and PCNA (S4 Fig) in diabetic rats, all of which were attenuated by DA-9701. Moreover, immunostaining further demonstrated altered expression patterns of c-Kit in the gastric tissues of diabetic rats (S5 Fig).

### Effects of DA-9701 on insulin secretion in STZ-induced diabetic and control groups

Insulin secretion was measured in both control and STZ-induced diabetic groups following glucose administration, with and without DA-9701 treatment. In the STZ-induced diabetic group, which exhibited reduced insulin secretion, DA-9701 treatment led to a significant increase in insulin levels at 30 minutes post-glucose administration compared to the STZ + Vehicle group (p < 0.05) (Fig 6).

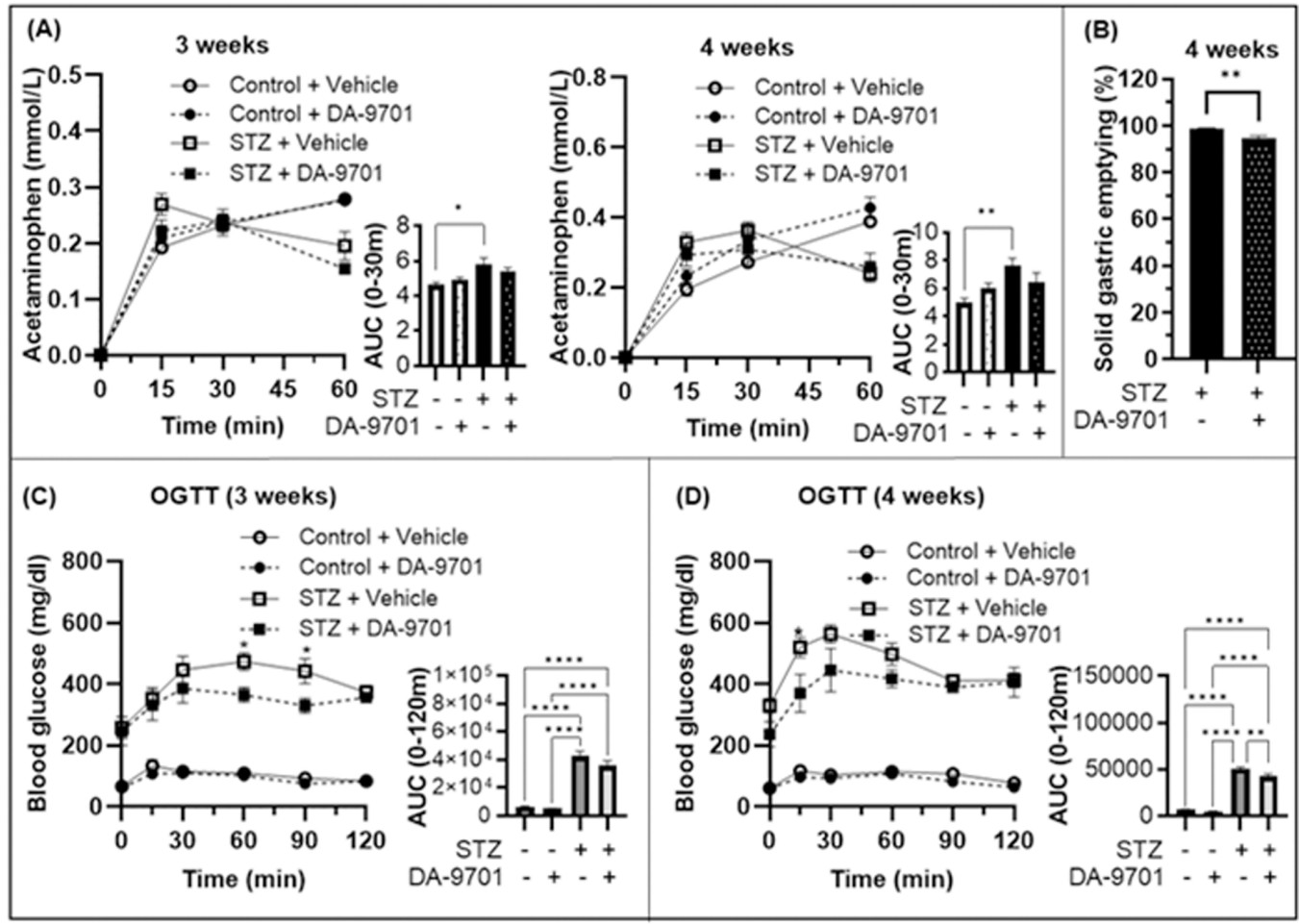

**Fig 4. Efficacy of DA-9701 in STZ-induced diabetic rats on gastric emptying and glucose levels.** (A) Liquid gastric emptying measured by plasma acetaminophen at three and 4 weeks post-STZ with and without DA-9701 treatment (n = 5/group). DA-9701 normalized emptying rates in STZ rats. (B) Comparison of solid gastric emptying in diabetic rats 2 weeks after vehicle or DA-9701 administration (n = 5/group). A marked increase in solid gastric emptying was observed in the STZ-only group, which was significantly modulated by the addition of DA-9701 (**p < 0.01). (C) Blood glucose levels from 0 minutes to 120 minutes after oral glucose administration (1 g/kg) in overnight-fasted rats, comparing the effects of DA-9701 and vehicle treatments in STZ-induced diabetic and control groups. The AUC for glucose variability highlighted the role of DA-9701 in reducing glucose spikes in STZ-treated rats (n = 5–10/group). (D) At 4 weeks, OGTT results demonstrated a significant effect of DA-9701 on glucose regulation in STZ-treated rats, with the AUC inset indicating improved glucose management.

## Discussion

Contrary to the long-standing view that diabetic gastroparesis is primarily associated with delayed gastric emptying, recent studies have revealed that gastric emptying can be significantly accelerated in the early stages of diabetes [2,7,8]. In this study, diabetic rats exhibited rapid gastric emptying from the onset of the disease. These findings suggest a dynamic and complex interaction between the progression of diabetes and gastrointestinal motility, underscoring the need to reassess the understanding of gastric emptying patterns in the context of diabetes. Such insights have important implications for redefining diagnostic and therapeutic approaches to gastrointestinal complications in patients with diabetes, particularly in its early stages.

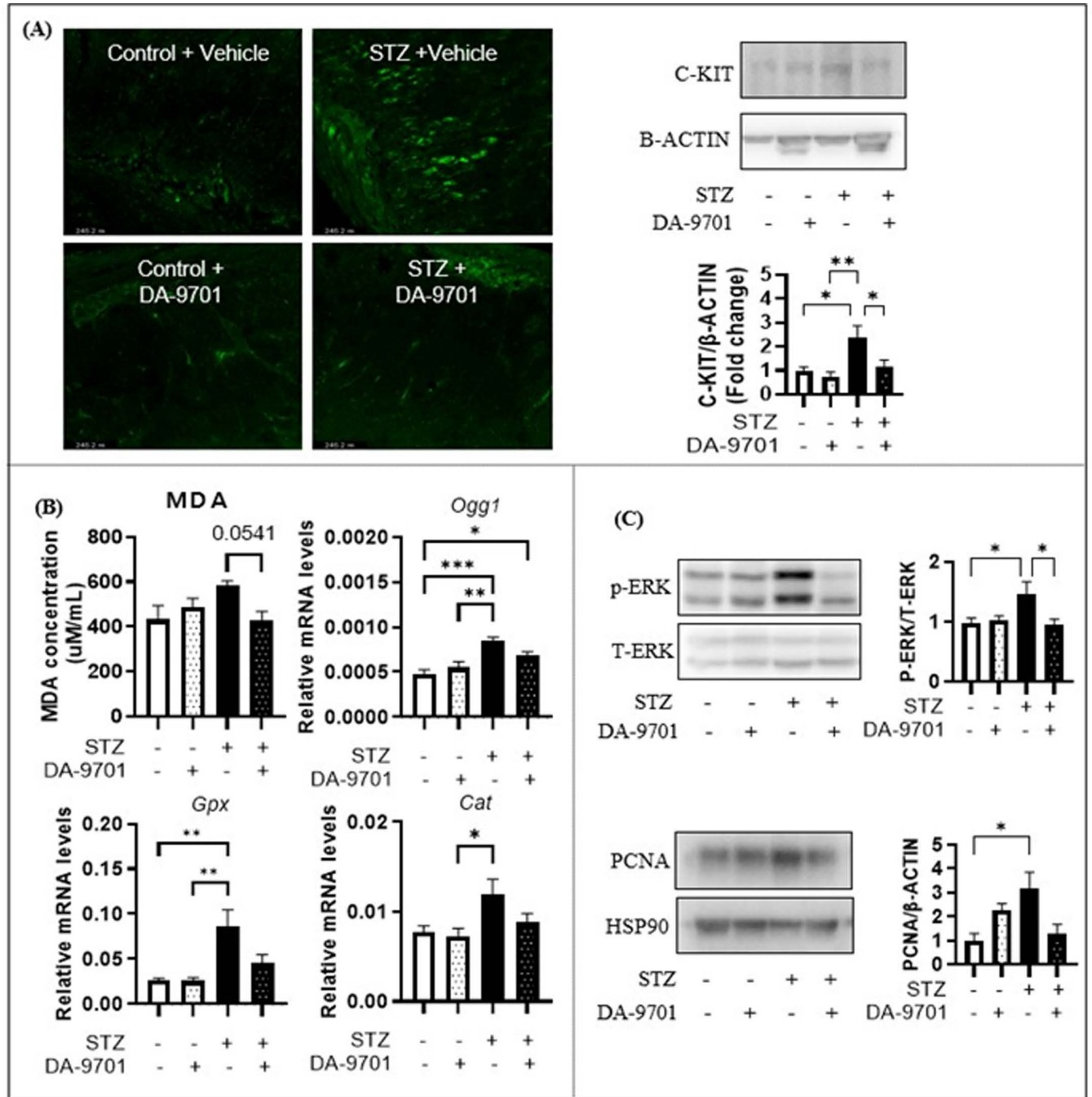

**Fig 5. Effects of DA-9701 on cellular and molecular markers in STZ-induced diabetic rats.** (A) Immnunostaining for c-Kit (green) and immunoblot analysis for c-Kit and β-Actin. STZ treatment upregulated c-Kit expression, which was significantly modulated by DA-9701 (*$p < 0.05$, **$p < 0.01$). (B) MDA concentration was measured to assess oxidative stress. STZ treatment resulted in elevated MDA levels, indicating increased oxidative stress; however, subsequent administration of DA-9701 mitigated this effect. The expression of oxidative stress-related genes (*Ogg1, Gpx, and Cat*) was measured using qRT-PCR. STZ treatment upregulated oxidative stress-related genes expression, which were significantly reduced by DA-9701 (*$p < 0.05$, **$p < 0.01$, ***$p < 0.001$). (C) Immunoblot analysis showed that expression of p-ERK1/2 and PCNA was upregulated in STZ-induced diabetic rats. All analyses were performed using gastric tissues from control and diabetic rats treated with vehicle or DA-9701 (6 mg/kg) daily for 2 weeks.

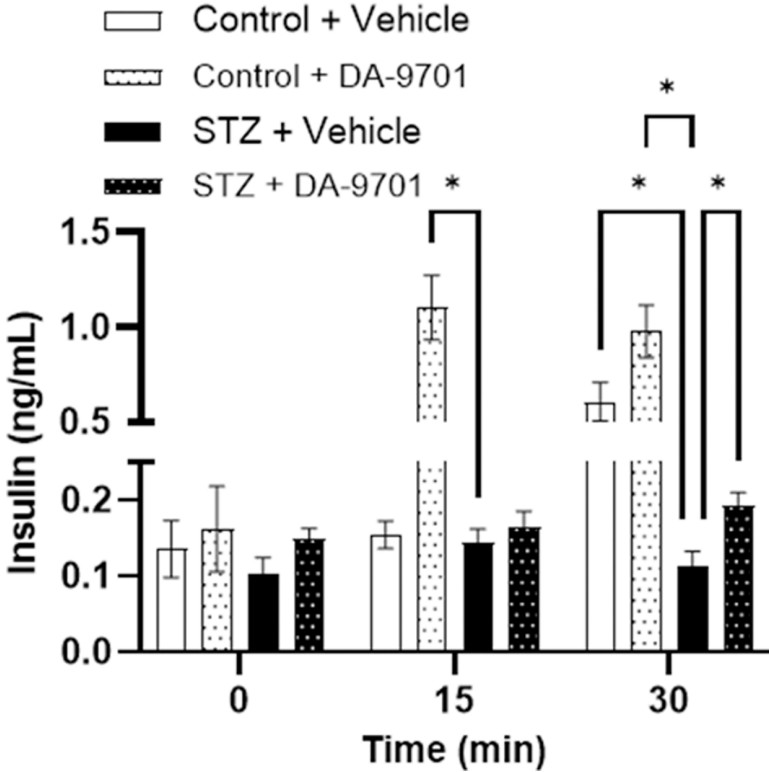

**Fig 6. Effects of DA-9701 on insulin secretion in STZ-induced diabetic group and control group.** Insulin levels (ng/mL) were measured at 0, 15, and 30 minutes post-glucose administration in control and STZ-induced diabetic groups treated with either vehicle or DA-9701. DA-9701 significantly increased insulin secretion at 15 and 30 minutes in both control and STZ-treated groups compared to their respective vehicle-treated groups (*$p < 0.05$).

This study focused on exploring the relationship between STZ-induced diabetes and alterations in the gastric emptying process, with particular attention to the role of interstitial cells of Cajal (ICCs). The central hypothesis was based on the notion that STZ-induced diabetes might lead to changes in the activity and proliferation of ICCs, which are crucial for coordinating gastric motility [1,2,14]. Therefore, the expression of c-Kit, a marker for ICCs, was measured in the gastric tissues of diabetic rats. The results demonstrated upregulated c-Kit expression, suggesting that ICC hyperplasia or increased ICC activity may occur in the early stages of diabetes. Such hyperplasia or increased activity of ICCs could be a significant contributing factor to the accelerated gastric emptying observed in these diabetic models [2,7,8]

The results emphasize the potential mechanistic link between diabetes-induced changes in gastric ICCs and the resulting alterations in gastric emptying, a notable aspect of gastrointestinal dysfunction in the context of diabetes [4,15].

We also measured MDA, and Ogg1 levels. MDA is a marker of oxidative stress, while Ogg1 plays a key role in DNA repair capacity and serves as a primary defense mechanism against oxidative stress. The increased levels of MDA and Ogg1 in the STZ-administered group indicated an elevation of oxidative stress in the diabetic state. Additionally, the increased activation of ERK1/2 and expression of c-Kit suggested alterations in signaling pathways associated with oxidative stress [7,8].

To further elucidate the oxidative stress response, we also performed quantitative real-time PCR analysis targeting key antioxidant defense genes, including glutathione peroxidase (Gpx) and catalase (Cat). The STZ group exhibited a compensatory upregulation of both Gpx and Cat mRNA levels, reflecting an endogenous attempt to counteract the heightened

oxidative burden [16]. In addition, we assessed the expression of proliferating cell nuclear antigen (PCNA), a marker of DNA replication and repair. PCNA mRNA expression was significantly increased in the STZ group, likely representing a cellular response to oxidative DNA damage and an attempt to maintain genomic integrity under diabetic conditions [17].

DA-9701 is a multi-targeted botanical drug used to treat functional dyspepsia in South Korea, with *Corydalis* tuber—the root of *Corydalis yanhusuo* (Papaveraceae)—as its main ingredient. This plant is known to regulate gastric acid secretion and prevent gastric and duodenal ulcers. *Corydalis* tuber extracts have been employed as antispasmodic agents and analgesics to alleviate abdominal pain due to their soothing and tranquilizing properties, and they have also demonstrated anti-inflammatory effects [9,10].

DA-9701 administration reduced oxidative stress by decreasing the levels of MDA and Ogg1, as well as attenuating the elevated expression of antioxidant defense genes Gpx and Cat, which are typically upregulated in response to oxidative burden. In parallel, the expression of proliferating cell nuclear antigen (PCNA), a marker of DNA replication and repair often induced by oxidative DNA damage, was also suppressed following DA-9701 treatment. This suggests that the reduction in oxidative stress limited the cellular need for DNA repair and proliferative compensation. The overall attenuation of oxidative stress was accompanied by decreased activation of ERK1/2 and c-Kit signaling pathways, thereby contributing to the normalization of the gastric emptying rate [7,8]. These results suggest that DA-9701 holds potential as a therapeutic agent for diabetes-related gastrointestinal abnormalities, as illustrated schematically in Fig 7.

Additionally, our study revealed that DA-9701 administration increased insulin secretion not only in the control group but also in STZ-induced diabetic rats [18,19]. The improvement in postprandial blood glucose levels observed with DA-9701 administration suggests that it could mitigate postprandial glucose spikes by normalizing gastric emptying [18,19]. This phenomenon indicates the potential of DA-9701 in regulating glycemic variability, which could be particularly beneficial in the early treatment of diabetes.

The changes in gastric emptying rates observed in diabetes may not solely result from alterations in cellular function but may also involve complex interactions among components of the gastrointestinal network. Furthermore, the increase in insulin secretion induced by DA-9701, observed for the first time in this study, necessitates further research to fully understand these dynamics.

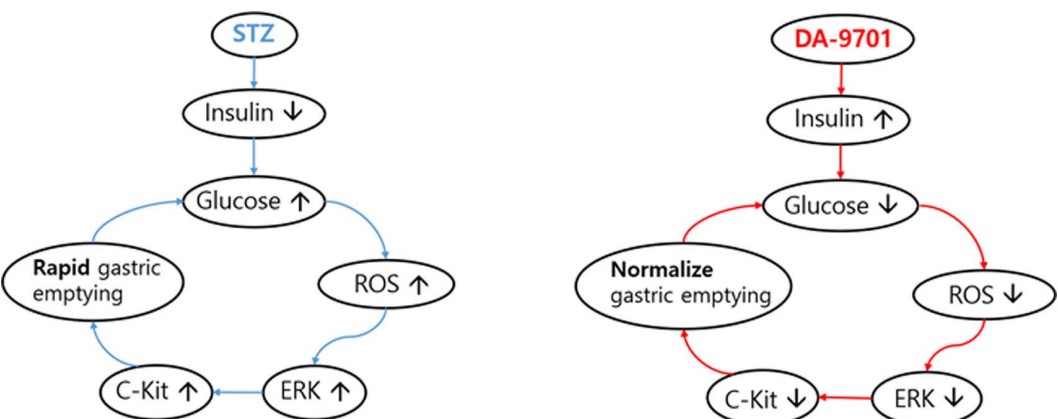

**Fig 7. A schematic showing a mechanism by which DA-9701 normalizes rapid gastric emptying.** Administration of STZ led to a series of events that induce hyperglycemia, which in turn increased oxidative stress, resulting in the activation of ERK1/2. This activation induced the upregulation of c-Kit, ultimately causing rapid gastric emptying. The administration of DA-9701 likely counteracted these effects, leading to the normalization of gastric emptying rates.

This study has elucidated the multifaceted nature of gastric motility in the early phase of diabetes, providing critical insights into the physiological complexities of this condition [1,4]. The results demonstrated that DA-9701 plays a crucial role in modulating gastric motility [11,12,20]. This compound effectively normalized the increased antral contractility, a key factor in the accelerated gastric emptying observed in early diabetes [1,4]. Moreover, the findings suggest that DA-9701 may also have a beneficial impact on insulin secretion, potentially offering dual therapeutic action in diabetes management.

This dual effect is particularly noteworthy, as it not only addresses the gastrointestinal symptoms commonly associated with diabetes but also aids in better control of blood glucose levels. Such a comprehensive approach to diabetes management is essential for enhancing patient care and outcomes.

However, this study had several limitations that warrant consideration. The duration was limited to 4 weeks post-diabetes induction, which may not fully capture the long-term effects of diabetes or the sustained impact of DA-9701. Thus, further research with longer observation periods is necessary. Additionally, the study utilized an STZ-induced diabetic rat model, which, although informative, may not directly translate to human diabetes, underscoring the need for clinical trials to validate these findings in humans.

Moreover, focusing on specific pathways such as ICCs, oxidative stress, and ERK1/2 signaling, while important, does not account for all factors influencing gastric motility, such as autonomic nervous system dysfunction, thereby limiting the comprehensiveness of the findings. Furthermore, although DA-9701 increased insulin secretion in diabetic rats, its applicability to human treatment remains uncertain and requires further investigation. The study also did not explore different dosages or administration routes, limiting the generalizability of the results. Lastly, although the sample size was adequate for statistical significance, it may still introduce variability, and larger studies under varied conditions are needed to corroborate these findings.

In conclusion, the study revealed that gastric emptying was significantly accelerated in the early stages of diabetes. This acceleration was effectively attenuated by DA-9701, which also contributed to the reduction of elevated blood glucose levels. These results offer a new perspective on diabetes management, emphasizing the importance of addressing both gastrointestinal symptoms and glycemic control to improve patient outcomes.

## Supporting information

**S1 File. Raw data for body weight, blood glucose, gastric emptying, oxidative stress markers, and Western blot analyses.**
(XLSX)

**S2 Fig. Western blot raw data for c-Kit.**
(TIF)

**S3 Fig. Western blot raw data for p-ERK.**
(TIF)

**S4 Fig. Western blot raw data for PCNA.**
(TIF)

**S5 Fig. Immunostain of c-Kit.**
(TIF)

## Author contributions

**Conceptualization:** Minjun Kim, Young Mi Song, Kyung Jae Yoon, Jung Ho Park, Yoo Kyung Kim, Chong Il Sohn.
**Data curation:** Minjun Kim, Young Mi Song, Yurim Hong, Yoo Kyung Kim, Juhee Kim, Chong Il Sohn.

**Formal analysis:** Minjun Kim, Young Mi Song, Yurim Hong, Juhee Kim, Chong Il Sohn.

**Investigation:** Minjun Kim, Chong Il Sohn.

**Methodology:** Minjun Kim, Young Mi Song, Yurim Hong, Jung Ho Park, Yoo Kyung Kim, Juhee Kim, Chong Il Sohn.

**Supervision:** Kyung Jae Yoon, Jung Ho Park.

**Validation:** Chong Il Sohn.

**Visualization:** Young Mi Song, Chong Il Sohn.

**Writing – original draft:** Minjun Kim, Young Mi Song.

**Writing – review & editing:** Young Mi Song, Kyung Jae Yoon, Jung Ho Park, Chong Il Sohn.

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
