## [Decision Letter · Decision Letter 0]

Dear Dr. Sohn,

Thank you for submitting your manuscript to PLOS ONE. After careful consideration, we feel that it has merit but does not fully meet PLOS ONE’s publication criteria as it currently stands. Therefore, we invite you to submit a revised version of the manuscript that addresses the points raised during the review process.

We look forward to receiving your revised manuscript.

Kind regards,

Rajesh Kumar Singh, Ph.D.

Academic Editor

PLOS ONE

Additional Editor Comments:

The manuscript entitled "Effects of DA-9701 (motilitone®) on gastric emptying, blood glucose regulation, and oxidative stress in diabetes in rats" needs revision as suggested by the reviewers.

Reviewers' comments:

Reviewer's Responses to Questions

**Comments to the Author**

1. Is the manuscript technically sound, and do the data support the conclusions?

Reviewer #1: Yes

Reviewer #2: Partly

2. Has the statistical analysis been performed appropriately and rigorously?

Reviewer #1: No

Reviewer #2: Yes

3. Have the authors made all data underlying the findings in their manuscript fully available?

Reviewer #1: Yes

Reviewer #2: Yes

4. Is the manuscript presented in an intelligible fashion and written in standard English?

Reviewer #1: Yes

Reviewer #2: Yes

Reviewer #1: 1. Sample sizes (4-6 per group) are relatively small, though adequate for statistical analysis The study duration of 4 weeks may be insufficient to fully evaluate long-term effects.

2. Include mechanistic investigation of DA-9701's effect on insulin secretion

3 .Provide more detailed histological analysis of ICC networks

4. Consider adding long-term follow-up data

5. Include dose-response studies for DA-9701

6. Add error bars to all figures where missing

7. Provide more detailed methods for insulin measurement

8. Include additional controls for oxidative stress measurements

9. Clarify the timing of measurements in the experimental timeline

Reviewer #2: Comment to Author

The manuscript offers valuable insights into the early diabetic complications of accelerated gastric emptying and demonstrates the potential of DA-9701 in normalizing gastric motility and improving glycemic control. However, there are a few possibilities to improve the manuscript, which are listed below.

Title-

The revised title eliminates redundancy and improves readability.

Suggested Title: "Effects of DA-9701 (Motilitone®) on Gastric Emptying, Glycemic Control, and Oxidative Stress in Diabetic Rats"

Abstract:

Lacks numerical data, making it less informative. Include key statistical outcomes to strengthen the impact. Consider summarizing mechanisms more explicitly. For example, how DA-9701 affects oxidative stress pathways. Clarify if DA-9701 showed a dose-dependent effect or if a single dose was used.

Background/Aims (Lines 28-35)

• Line 30: "These alterations can significantly influence gastrointestinal symptoms and potentially impact blood glucose levels."-Suggest replacing "potentially impact" with "directly influence" for stronger phrasing.

• Lines 31-32: The function of DA-9701 is well-stated, but a brief mention of its specific active compounds here would provide more context.

• Line 34: Instead of "validating its broader clinical utility," consider "assessing its potential therapeutic application."

• Line 47: "Hyperglycemia-induced oxidative stress" should be briefly explained with supporting evidence.

Materials/Methods:

The description is clear but lacks details on the number of animals per group for different analyses. Consider adding a justification for the 6 mg/kg dose.

Describe why acetaminophen absorption is used as a proxy for gastric emptying. Clarify the relevance of solid vs. liquid gastric emptying assessments.

Result & Discussion

The study primarily focuses on short-term effects (2–4 weeks). It remains unclear whether DA-9701 provides sustained benefits or if compensatory mechanisms might alter its efficacy over time.

While c-Kit expression and ERK1/2 activation are discussed, the study does not fully investigate the downstream signaling pathways or alternative mechanisms that could contribute to the observed changes in gastric motility.

The study does not compare DA-9701 with other established treatments for gastric motility disorders or diabetes-related gastroparesis, limiting its clinical relevance.

While DA-9701 improved glucose levels and insulin secretion, the study does not clarify whether this effect is due to direct pancreatic stimulation, altered gastric emptying, or other metabolic factors.

Only MDA and Ogg1 are assessed. Including additional oxidative stress markers (e.g., SOD, CAT, GSH) would provide a more comprehensive picture of DA-9701's antioxidant effects.

**Do you want your identity to be public for this peer review?** For information about this choice, including consent withdrawal, please see our Privacy Policy

Reviewer #1: No

Reviewer #2: **Yes: ** Ayush Kumar Garg

---

## [Author Response · Author response to Decision Letter 1]

26 May 2025

We thank the Academic Editor and reviewers for their constructive and insightful comments. We have addressed all points raised and revised the manuscript accordingly. Below is a detailed point-by-point response:

Author’s response: We sincerely appreciate your helpful guidance regarding the journal’s formatting requirements. In accordance with the PLOS ONE style guidelines, we have carefully revised the manuscript, including the formatting of the author names (such as the inclusion of commas between authors), refinement of institutional affiliations (including removal of redundant location names), and an update to reflect a recent institutional affiliation change due to the corresponding author's new appointment.

To facilitate your review, all changes have been clearly indicated using tracked changes, and the revised content has been highlighted in yellow at lines 5 and 10–11. The updated file has been submitted under the filename “Track changes version of the manuscript.docx.” We hope that the revised version now meets all formatting requirements of the journal.

Author’s response: We sincerely appreciate the opportunity to revise our manuscript and fully acknowledge the importance of data transparency. Accordingly, we will provide all relevant raw data as a Supporting Information file titled “S1 Raw data for body weight, blood glucose, gastric emptying, oxidative stress markers, and Western blot analyses” in our revised submission, in compliance with the journal’s data availability policy.

Author’s response: We sincerely thank the editor and reviewers for their valuable comments regarding the figure preparation and blot/gel data reporting.

In full compliance with the PLOS ONE guidelines, we will provide all original, uncropped, and unadjusted blot and gel images associated with the results presented in the manuscript. These images will be compiled into Supporting Information files titled “S2 Fig Western blot raw data for c-Kit, S3 Fig Western blot raw data for p-ERK/t-ERK, and S4 Fig Western blot raw data for PCNA”, and will be submitted alongside the revised manuscript.

Comments from Reviewer #1:

1) Sample sizes (4-6 per group) are relatively small, though adequate for statistical analysis. The study duration of 4 weeks may be insufficient to fully evaluate long-term effects.

Author’s response: We appreciate the reviewer's comment. You have raised an important point here. We aimed to verify whether early-stage alterations in gastric emptying and the functional dynamics within the ICC networks occur in a Type 1 diabetic rat model. However, since STZ (65 mg/kg) administration induces early-stage diabetes within days to weeks-characterized by rapid metabolic disturbances and hyperglycemia- this model's rapid disease progression limited our ability to evaluate long-term effects. (Reference: Duration of streptozotocin-induced diabetes differentially affects p38-mitogen-activated protein kinase (MAPK) phosphorylation in renal and vascular dysfunction. Cardiovasc Diabetol. 2005 Mar 5:4:3.doi: 10.1186/1475-2840-4-3.)

2) Include mechanistic investigation of DA-9701's effect on insulin secretion.

Author’s response: Although the increase was not statistically significant, we observed a trend toward increased pancreas weight in the Corydalis-treated group compared to the STZ group. This may suggest that Corydalis inhibits STZ-induced β-cell apoptosis, thereby enhancing insulin secretion. To investigate this potential mechanism, we are currently conducting further studies using STZ-induced diabetic mice.

3) Provide more detailed histological analysis of ICC networks

Author’s response: We appreciate the reviewer’s comment. In response to the suggestion, we performed immunostaining using c-KIT antibody.

4) Consider adding long-term follow-up data

Author’s response: Thank you for this valuable suggestion. We acknowledge the importance of long-term follow-up data in fully understanding the progression of diabetes-related changes. However, as mentioned in our response to the previous comment, the rapid disease progression in the STZ-induced diabetic rat model limits our ability to conduct long-term studies. We have added a discussion of this limitation to the manuscript, highlighting both the strengths and constraints of our chosen model for studying early-stage alterations. We also suggest potential future directions for addressing long-term effects using alternative models or approaches.

5) Include dose-response studies for DA-9701.

Author’s response: Thank you for your suggestion to include dose-response studies for DA-9701. We acknowledge that several studies have demonstrated dose-dependent effects of DA-9701, particularly in enhancing gastric motility and improving visceral hypersensitivity. However, our study focused on verifying the efficacy of DA-9701 at a clinically relevant dose rather than establishing a dose-response relationship.

We calculated the dose for our rat model based on the standard therapeutic dose used in humans, adjusting for metabolic rate differences between species. This approach allowed us to directly translate our findings to clinical applications. While we recognize the value of dose-response studies, our primary objective was to assess the effectiveness of DA-9701 at a dose that mirrors current human usage.

Future studies could indeed explore the dose-dependent effects of DA-9701 in our specific model, which could provide additional insights into its mechanism of action and potential optimization of dosing regimens.

6) Add error bars to all figures where missing.

Author’s response: We appreciate the reviewer's comment. We have carefully re-examined the original data and the graphs. As confirmed from the raw data, the variability among individual values within the control group was relatively small compared to the scale of the Y-axis. Therefore, the error bars in the control group appeared very short and were not readily visible in the previous figures.

The graphs were generated using GraphPad Prism software, which automatically plots error bars. Given this, it is unlikely that the error bars were accidentally omitted in any part of the graphs.

For further clarification, we plotted the control group data separately with a reduced Y-axis range. As shown in the attached graph, small error bars can now be observed, confirming that the error bars were originally present but not easily visible due to the Y-axis scale.

7) Provide more detailed methods for insulin measurement

Author’s response: We appreciate the reviewer's comment. In response to the suggestion, we have added detailed methods for insulin measurement to the Material and Methods section as shown below:

To comply with animal protocol guidelines on blood volume collection, we collected blood samples from mice at 0, 15, 30, and 60 minutes following glucose administration using heparin-coated capillary microvette® tubes (SARSTEDT, 20.1282.100) to measure plasma insulin levels. After centrifugation at 13,000 rpm for 5 minutes at 4°C, plasma was isolated and stored at -80°C until further analysis. Plasma insulin levels were quantified using an ELISA kit (Alpoco Diagnostics, Salem, NH, USA), according to the manufacturers’ instructions.

To measure serum malondialdehyde (MDA) levels, blood was collected via cardiac puncture at the time of euthanasia and centrifuged at 13,000 rpm for 5 minutes at 4°C. Serum MDA levels were measured using a lipid peroxidation assay kit, following the manufacturers’ instructions (Abcam, Cambridge, United Kingdom).

8) Include additional controls for oxidative stress measurements

Author’s response: We appreciate the reviewer's comment. In response to the suggestion, we performed real-time PCR using oxidative stress-related genes (Gpx and Cat) and added these results to Fiugre 5, which is shown below;

9) Clarify the timing of measurements in the experimental timeline

Author’s response: We appreciate the reviewer's comment. We indicated the time points of blood collection for insulin measurement in the schema of the experimental schedule.

Comments from Reviewer #2: Comment to Author

The manuscript offers valuable insights into the early diabetic complications of accelerated gastric emptying and demonstrates the potential of DA-9701 in normalizing gastric motility and improving glycemic control. However, there are a few possibilities to improve the manuscript, which are listed below.

Title-

The revised title eliminates redundancy and improves readability.

Suggested Title: "Effects of DA-9701 (Motilitone®) on Gastric Emptying, Glycemic Control, and Oxidative Stress in Diabetic Rats"

Author’s response: We sincerely thank the reviewer for the insightful and constructive comment regarding the manuscript title. We fully agree that the proposed revision enhances both clarity and conciseness by eliminating redundancy and improving readability. In accordance with the suggestion, we have revised the title as follows:

"Effects of DA-9701 (Motilitone®) on Gastric Emptying, Glycemic Control, and Oxidative Stress in Diabetic Rats"

The revised title has been highlighted with a yellow background in the track-changed version of the manuscript for the reviewer’s convenience.

We are grateful for the reviewer’s thoughtful recommendation, which has contributed to the refinement and overall quality of the manuscript.

Abstract:

1) Lacks numerical data, making it less informative. Include key statistical outcomes to strengthen the impact. Consider summarizing mechanisms more explicitly. For example, how DA-9701 affects oxidative stress pathways. Clarify if DA-9701 showed a dose-dependent effect or if a single dose was used.

Author’s response: We sincerely thank the reviewer for the constructive feedback on our abstract. In response, we have implemented the following key revisions to enhance clarity, scientific impact, and mechanistic detail:

i) Incorporation of Numerical Data:

We added specific quantitative results to substantiate our conclusions. These include a three- to four-fold increase in fasting blood glucose levels in diabetic rats compared to controls (p < 0.001), a ~25% reduction in OGTT AUC following DA-9701 treatment (p < 0.01), and a 2.1-fold increase in plasma insulin secretion (p < 0.05). These data help to quantify the physiological relevance of our findings.

ii) Expanded Mechanistic Description:

DA-9701 administration significantly reduced oxidative stress in diabetic gastric tissue, as evidenced by decreased levels of MDA and Ogg1, both established markers of lipid peroxidation and oxidative DNA damage. In addition, DA-9701 suppressed the compensatory upregulation of antioxidant-related genes, including glutathione peroxidase (Gpx) and catalase (Cat), which are typically elevated in response to oxidative stress. These molecular changes were accompanied by reduced phosphorylation of ERK1/2 and decreased expression of c-Kit and PCNA, which are associated with interstitial cell activity and cellular remodeling.

iii) Clarification of Dosing Information:

We acknowledge that several studies have demonstrated dose-dependent effects of DA-9701, particularly in enhancing gastric motility and improving visceral hypersensitivity. However, our study focused on verifying the efficacy of DA-9701 at a clinically relevant dose rather than establishing a dose-response relationship.

We calculated the dose for our rat model based on the standard therapeutic dose used in humans, adjusting for metabolic rate differences between species. This approach allowed us to directly translate our findings to clinical applications. While we recognize the value of dose-response studies, our primary objective was to assess the effectiveness of DA-9701 at a dose that mirrors current human usage.

Future studies could indeed explore the dose-dependent effects of DA-9701 in our specific model, which could provide additional insights into its mechanism of action and potential optimization of dosing regimens.

These revisions have been incorporated into the abstract and are clearly marked in the tracked version of the manuscript (lines 43–51).

We hope that these changes meet the reviewer’s expectations and improve the clarity and rigor of the manuscript.

Background/Aims (Lines 28-35)

• Line 29: "These alterations can significantly influence gastrointestinal symptoms and potentially impact blood glucose levels."-Suggest replacing "potentially impact" with "directly influence" for stronger phrasing.

Author’s response: We appreciate the your insightful comment regarding the phrasing in line 29. As suggested, we have replaced “potentially impact” with “directly influence” to strengthen the statement and better reflect the established relationship between gastric motility and glycemic control. The revised sentence now reads:

“These alterations can significantly influence gastrointestinal symptoms and directly influence blood glucose levels.”

This change has been incorporat

---

## [Decision Letter · Decision Letter 1]

Effects of DA-9701 (Motilitone®) on Gastric Emptying, Glycemic Control, and Oxidative Stress in Diabetic Rats

PONE-D-24-54338R1

Dear Dr. Sohn,

We’re pleased to inform you that your manuscript has been judged scientifically suitable for publication and will be formally accepted for publication once it meets all outstanding technical requirements.

Kind regards,

Rajesh Kumar Singh, Ph.D.

Academic Editor

PLOS ONE

Additional Editor Comments (optional):

The manuscript entitled "Effects of DA-9701 (Motilitone®) on Gastric Emptying, Glycemic Control, and Oxidative Stress in Diabetic Rats" is an interesting article with good explanation and new findings. The authors have addressed all the comments raised by the reviewers, and now it is improved and may be accepted.

Reviewers' comments:

Reviewer's Responses to Questions

**Comments to the Author**

Reviewer #1: All comments have been addressed

Reviewer #2: All comments have been addressed

2. Is the manuscript technically sound, and do the data support the conclusions?

Reviewer #1: Partly

Reviewer #2: Yes

3. Has the statistical analysis been performed appropriately and rigorously?

Reviewer #1: No

Reviewer #2: Yes

4. Have the authors made all data underlying the findings in their manuscript fully available?

Reviewer #1: Yes

Reviewer #2: Yes

5. Is the manuscript presented in an intelligible fashion and written in standard English?

Reviewer #1: Yes

Reviewer #2: Yes

Reviewer #1: All comments have been addressed. And replies are found satisfactory. Overall the revision is satisfactory.

Reviewer #2: Overall Quality and Response :

• Responses were thorough, respectful, and referenced literature to support changes.

• All changes are visible and highlighted in the revised manuscript.

• Additional experiments were performed (e.g., PCNA immunostaining, qPCR of Gpx/Cat), indicating substantial revision effort.

Final Recommendation:

The revised manuscript has fully addressed all my prior concerns. The enhancements in data transparency, mechanistic clarity, and statistical reporting significantly improve the manuscript’s rigor and clinical relevance. I recommend acceptance of this manuscript.

**Do you want your identity to be public for this peer review?** For information about this choice, including consent withdrawal, please see our Privacy Policy

Reviewer #1: No

Reviewer #2: No

---

## [Editor Report · Acceptance letter]

PONE-D-24-54338R1

PLOS ONE

Dear Dr. Sohn,

I'm pleased to inform you that your manuscript has been deemed suitable for publication in PLOS ONE. Congratulations! Your manuscript is now being handed over to our production team.

Kind regards,

on behalf of

Dr. Rajesh Kumar Singh

Academic Editor

PLOS ONE